# Effects of Stocking Density and Pre-Slaughter Handling on the Fillet Quality of Largemouth Bass (*Micropterus salmoides*): Implications for Fish Welfare

**DOI:** 10.3390/foods13101477

**Published:** 2024-05-10

**Authors:** Nima Hematyar, Samad Rahimnejad, Swapnil Gorakh Waghmare, Oleksandr Malinovskyi, Tomas Policar

**Affiliations:** 1Faculty of Fisheries and Protection of Waters, South Bohemian Research Center of Aquaculture and Biodiversity of Hydrocenoses, Research Institute of Fish Culture and Hydrobiology, University of South Bohemia in Ceske Budejovice, Zátiší 728/II, 389 25 Vodňany, Czech Republic; s.rahimnejad@um.es (S.R.); swaghmare@mcw.edu (S.G.W.); omalinovskyi@frov.jcu.cz (O.M.); policar@frov.jcu.cz (T.P.); 2Immunobiology for Aquaculture Group, Department of Cell Biology and Histology, Faculty of Biology, Regional Campus of International Excellence “Campus Mare Nostrum”, University of Murcia, 30100 Murcia, Spain

**Keywords:** welfare, anoxia, oxidation development, blood biochemistry, antioxidant capacity

## Abstract

There is currently insufficient acknowledgment of the relationship between fish welfare and ultimate fillet quality. The purpose of this study was to assess the impacts of pre-slaughter handling and stocking density as fish welfare markers on fillet quality of largemouth bass (*Micropterus salmoides*). Fish from three stocking densities of 35, 50, and 65 kg·m^−3^ were reared in a recirculating aquaculture system (RAS) for 12 weeks and received commercial feed. Ultimately, the fish were either stunned with percussion on the head (control group) or subjected to air exposure for 3 min (anoxia group) before stunning and subsequent collection of blood and fillet samples. Western blot analysis revealed the degradation of actin in both groups. Additionally, higher oxidation progress and lower hardness and pH were observed in anoxia compared to the control group. We observed higher hardness at 35 kg·m^−3^ in anoxia compared to 50 and 65 km^−3^. The initial hardness values at 35, 50, and 65 km^−3^ were 1073, 841, and 813 (g) respectively in the anoxia group. Furthermore, the anoxia and control groups had rigor mortis after 6 and 10 h, respectively. Cortisol and glucose levels, and oxidative enzymes activity were higher in anoxia than in the control group. In conclusion, oxidation induced by anoxia likely plays a crucial role as a promoter of the quality deterioration of largemouth bass fillets.

## 1. Introduction

Largemouth bass (*Micropterus salmoides*) is one of the fish species well-suited for aquaculture production [1]. Due to its exceptional fillet quality, willingness to consume commercial feed, and rapid growth rate, it is a good fit for intensive aquaculture [2,3].

Seafood products quality can be assessed by various criteria such as nutritional quality, food safety standards, and sensory analysis [4]. The freshness of fish fillet is influenced by different processing conditions, a crucial aspect for the fishing industry.

Throughout pre-slaughter handling, animals may undergo physical and psychological stressors, leading to heightened anxiety, increased heart rate, respiration, and irritability. Overconsumption of nutrients and water triggers hormonal secretion, resulting in changes to biochemical indicators in the plasma and energy metabolism, adversely impacting animal welfare, production yield, and meat quality [5,6]. There has been a growing concern for animal welfare like stunning methods and stocking density, which are now recognized as critical elements for ensuring food safety and fillet quality. The interval preceding slaughter, during which fish are removed from the water, induces stress in the fish. Inadequate slaughter methods can either expedite or prolong the resolution of rigor mortis in fish, contingent upon their stress level. The 3-min anoxia approach aligns with the guidelines of [7]. The stocking density during aquaculture and slaughter procedures are key aspects of fish welfare, which could also affect the quality of fish fillets. However, assessing fish welfare proves challenging given the lack of consensus regarding pain receptors and anxiety in fish. Additionally, there is insufficient research aimed at establishing a correlation between fillet quality and fish welfare [8]. Furthermore, factors such as handling and rearing conditions including transportation, water temperature and quality, and stocking density serve as indicators of fish welfare, which in turn affect the quality of fish fillets [9]. Increasing stocking density can enhance fish productivity; however, it is associated with reduced fish welfare. Given the influence of stocking density on fillet quality, assessing fillet quality can offer a valuable understanding of both fish welfare and the economic conditions of fish rearing, thereby enhancing the quality of the final product.

Poor slaughtering procedures can elevate stress levels in fish, hastening the onset of rigor mortis [10]. The stress that animals undergo before slaughter can elevate lactic acid production and diminish water-holding capacity (WHC), promoting protein degradation, lipid oxidation, and microbial growth [11]. The rapid decline in pH after slaughtering causes the denaturation of myosin protein and alters its solubility to an insoluble state.

Seafood products are highly susceptible to microbial and enzymatic activity during postmortem conditions [12]. The quality of seafood products may be negatively impacted by protein and lipid oxidation, as along with endogenous enzymatic activity, leading to a decrease in their value [13]. Previous studies have examined the effect of slaughtering stress on the onset of lipid oxidation in African catfish (*Clarias gariepinus*) and gilthead seabream (*Sparus aurata*) [14,15]. In marine organisms, various environmental stressors trigger the generation of reactive oxygen species (ROS); with stressors stimulating their production. This can lead to an enhanced antioxidant capacity in fish when exposed to abnormal culture conditions. The in-vivo systems possess cellular defense mechanisms such as glutathione peroxidase (GPx) and superoxide dismutase (SOD) to remove the accumulation of ROS. For instance, glutathione peroxidase catalyzes peroxide reduction by oxidizing glutathione (GSH) to oxidized glutathione (GSSG).

The purpose of this study was to investigate the effects of fish welfare indices, such as stocking density and pre-slaughter procedures, on the quality of fillet and protein profile of largemouth bass obtained from fish intensively reared under different constant stocking densities. Additionally, the study aimed to establish a relationship between pre-slaughter conditions and fillet quality.

## 2. Materials and Methods

### 2.1. Ethical Statement

The University of South Bohemia authorized all animal procedures and licenses were provided to Nos. 58672/2020-MZE-18134 and Nos. 33446/2020-MZE-18134 under the NAZV QK22020144 project.

### 2.2. Fish Rearing under Different Densities

Largemouth bass (*Micropterus salmoides*) juveniles with a total length (TL) of 35–45 mm and body weight (BW) of 0.37–0.42 g was obtained from the Fishery School in the Czech Republic. These fish were harvested from a traditional pond used for juvenile fish production in early July. The fish were acclimated to the recirculating aquaculture system conditions for four weeks and fed on a commercial feed (Europa 15, 2–3 mm size pellets, Skretting) according to [16]. Also, the optimal environmental conditions described by [3] were applied. The water temperature during the experiment was 25.5 ± 0.8 °C. The experimental fish (TL 235 ± 21.7 mm and BW 163 ± 55.4 g) were reared in the same RAS under three different constant densities (D) of 35 (D35), 50 (D50), and 65 (D65) kg·m^−3^, presenting three experimental groups. The fish were cultured in a total of nine tanks (each density group had three replications) for three months to reach a marketable size of TL = 388 ± 39.7 mm and BW = 351 ± 72.3 g. Prior to sampling, the fish underwent a fasting period and were given a 48-h purification period in tanks with a temperature of 26 °C and 96% oxygen saturation.

### 2.3. Slaughtering Method

After fasting, 54 experimental fish from each density group (18 fish per replicate) were randomly captured and subjected to an anoxia test by keeping the fish out of water for 3 min [7]. In this way, three experimental anoxia (AN) groups were created: D35AN, D50AN, and D65AN. The fish were subjected to acute antemortem stress through anoxia and percussion on the head, a common practice in fish farming before slaughter. The same number of fish (*n* = 54) from each stocking density group were killed by direct percussion on the head without air exposure. These three groups were considered control (C) groups (D35C, D50C, and D65C). All the groups (D35AN; D50AN; D65AN; D35C; D50C, and D65C) were handled under the same conditions to avoid additional stress.

### 2.4. Sample Collection

After slaughter, the fish were cleaned in cold water and divided into 108 fillets, with two fillets from each fish, resulting in 54 right and 54 left fillets per group. The fillets collected from each fish were individually packed in plastic bags to avoid drying the fillet surface and stored at 4 °C. Fillet quality was assessed immediately and after 24 and 48 h of refrigerator storage as follows. In total, six left-hand fillets were randomly selected from each experimental group for hardness analysis. On the other hand, six randomly selected right-hand fillets were used for TBARS and proteomics analysis. For analysis of antioxidant capacity (catalase, superoxide dismutase, and glutathione peroxidase), three right-hand fillets were taken from each group immediately after slaughter. All flesh samples were frozen with liquid nitrogen and kept at −80 °C until analysis [17]. 

In addition, in total, 162 whole fish (27 fish per group) were sampled separately to determine the rigor index and pH. As part of the experiment, three fish were randomly chosen from each group and placed on ice for various durations, namely 0, 6, 9, 12, 24, 36, 48, 60, and 72 h. In this regard, we used the same fish for each time point to measure the pH and rigor index. 

After administering anesthesia with 0.03 mL/L clove oil, blood samples were collected from the caudal vein of 18 fish (3 from each group) right after they were slaughtered and before they were filleted. Heparinized syringes were used for this procedure. The plasma samples were used for evaluation of the stress response factors such as glucose and cortisol concentrations.

### 2.5. Analytical Procedures

#### 2.5.1. pH

The pH levels of fish were measured at different time intervals (0, 6, 9, 12, 24, 36, 48, 60, and 72 h) using a pH probe (Testo 206, Lenzkirch, Germany). Three fish per group were tested and the probe was placed behind the head into the higher mass of the fillet. The statistical bars were not included in this graph to ignore complications. However, the *p*-value was added as a Appendix A.

#### 2.5.2. Rigor Index

Rigor mortis development was assessed using the tail drop method [16] using 3 whole fish from each group during 72 h of post slaughtering at intervals of 0, 6, 9, 12, 24, 36, 48, 60, and 72 h. The entire fish was stored on ice while measuring the rigor index. The rigor index (Ir) was determined using the following formula: Ir = [(Lo − Lt)/Lo] × 100. Here, L is the vertical drop (in cm) of the tail when half of the fish fork length was placed on the edge of a table. This measurement was taken as a function of time. The initial tail drop is Lo, while measurements throughout the experiment (t = 0–72 h) are represented by Lt. T = 0, indicating the time immediately after slaughter. Statistical bars were excluded to avoid complications. However, the *p*-value was included as a Appendix A.

#### 2.5.3. Analysis of Glucose and Cortisol Concentrations

The blood samples collected for biochemical analysis were placed in a microcentrifuge (MPW 55, MPW Instruments, Poland, Ohio) and centrifuged at 1500× *g* for 10 min. The resulting plasma was separated and transferred to Eppendorf tubes, which were kept on ice. The tubes were then stored at −80 °C for subsequent analysis of glucose (GLU) and cortisol (COR) concentrations using a VETTEST 8008 hematology analyzer (IDEXX Laboratories Inc., Westbrook, ME, USA).

#### 2.5.4. Antioxidant Capacity

The catalase activity (CAT; EC 1.11.1.6) was measured by spectrophotometric analysis of H_2_O_2_ decomposition at 240 nm (*n* = 6) [18]. The total activity of superoxide dismutase (SOD; EC 1.15.1.1) was measured using a spectrophotometer at a wavelength of 560 nm as described by [19]. The measurement of glutathione peroxidase (GPX; EC 1.11.1.9) involved estimating the reduction in nicotinamide adenine dinucleotide phosphate (NADPH) over time by using reduced glutathione (GSH) as a co-substrate. This was further converted to glutathione disulfide (GSSG) to become GPX. The reduction rate of NADPH was measured by its absorbance at 340 nm [20].

#### 2.5.5. Hardness Analysis

The instrumental hardness analysis was conducted on six left fillets from each group using a texture analyzer called TA-XT Plus, manufactured by Stable Micro Systems, Godalming, UK. The fillet was flattened to half of its initial thickness by using a flat-ended cylinder (10 mm diameter, P/10 type). The cylinder was applied perpendicular to the muscle fibers, below the dorsal fin. The pressure was applied at a speed of 2 mm/s. Hardness was defined based on the greatest force measured during the first compression, represented in grams. To avoid confusion, the statistical bars have been excluded from the graph. However, the *p*-value was added as a Appendix A.

#### 2.5.6. Thiobarbituric Acid Reactive Substances (TBARS)

The measurement of lipid oxidation was conducted through the thiobarbituric acid reactive substances (TBARS) method, according to [21]. Six right fillets in each group were used for the semi-frozen samples. These samples were chopped, and connective tissues and detectable fat were eliminated. Approximately 1 g of muscle tissue was extracted from each subsample by using an Ultra Turrax from Janke and Kunkel, Staufen, Germany. Then, the samples were homogenized with 9.1 mL (0.61 mol·L^−1^) of trichloroacetic acid (TCA) solution and 0.2 mL (0.09 mol·L^−1^) of butylated hydroxytoluene (BHT) in methanol at a speed of approximately 14,000 rpm for 3 sets of 20 s each. The homogenate was passed through Munktell paper (Munktell Filter AB, Grycksbo, Sweden) and filtered. Then, two times, 1.5 mL of the filtrate was moved to new tubes. To the first sample (test sample), 1.5 mL of a solution containing 0.02 mol/L of thiobarbituric acid (TBA) was added, while to the second sample (sample blank), 1.5 mL of water was added. The reaction was allowed to proceed in darkness for 15–20 h at room temperature (20 °C). After that, the reaction complex was detected using a UV–visual spectrophotometer (Specord 210; Analytik, Jena, Germany) at a wavelength of 530 nm, against the sample blank. The amount of TBARS was expressed in terms of malondialdehyde (MDA) (µg/g). In this graph, statistical bars were not included to avoid complications. However, the *p*-value was added as a Appendix A.

#### 2.5.7. Proteomics Sample Preparation

##### Muscle Protein Extraction

To minimize protein degradation, almost 100 mg of frozen fish muscle from six right fillets in each group was cut and weighed at −20 °C. The muscle tissue was ground in 50 mM phosphate buffered saline (PBS) solution (500 µL). This solution was prepared with a 0.01 M phosphate buffer concentration and a 0.154 M sodium chloride concentration at pH 7.4. The crude extracts were then transferred to an Eppendorf tube.

##### SDS–PAGE

Standard protocols conducted sodium dodecyl sulfate-polyacrylamide gel electrophoresis (SDS-PAGE) [22]. For each sample, 20 µL of the solution was mixed with Laemmli sample buffer to attain a final protein concentration of 2 µg·µL^−1^. The mixture was then heated for 2 min at a temperature of 95 °C. After heating, the samples were loaded onto a 10% Criterion Tris-glycine Gel from Bio-Rad, located in Hercules, CA, USA. The samples were then subjected to electrophoresis by applying a constant electrical potential of 200 V. The protein ladder utilized was the Spectra Multicolor Broad Range (15–220 kDa) from Thermo Scientific, Rockford, IL, USA. The gel was then electrophoresed and stained with 0.5% Coomassie Brilliant Blue G-250 (Bull Korean Chem Soc. 2002, Eschenstr, Germany).

##### Immunoblotting

For immunoblotting, the DNPH reaction was performed directly on the protein homogenate obtained by centrifugation at 12,600× *g* for 3 min. The supernatant was utilized for analysis after adjusting the protein concentration to 10 mg/mL using a BCA kit from Pierce, Rockford, IL. Protein carbonyls were treated with a mixture of 20 µL of sample (1:2), 12% SDS, 10% TFA, and 10 mM DNPH and incubated at room temperature for 30 min. The reaction was halted by adding 40 µL of neutralization buffer (1:2) containing 2 M Tris-base, 30% glycerol, and 20 mM dithioerythritol (DTE) before being separated through SDS-PAGE. The experiment involved centrifuging the samples for 3 min at a force of 12,600× *g*, followed by loading them onto a 10% Tris-glycine gel (Bio-Rad, Hercules, CA, USA) for SDS-PAGE. The gels were then transferred onto 0.2 µm polyvinylidene difluoride (PVDF) membranes using a Trans-Blot SD semidry transfer cell (Bio-Rad Laboratories, Hercules, CA, USA) at 0.35 A and max 50 V for 60 min. The next step involved blocking the membranes with 5% skim milk in Tris-buffered saline and incubating them overnight at +4 °C with anti-DNP (Sigma Aldrich, Darmstadt, Germany), produced in rabbit, at a 1:16,000 dilution in Tris-buffered saline (TBS). After being washed in TBS, the membranes were incubated with the secondary antibody, horseradish peroxidase-conjugated swine anti-rabbit (DAKO Denmark A/S, Glostrup, Denmark), at a 1:8000 dilution, followed by another wash in TBS. Finally, the blot was developed using an ECL ± kit (Bio-Rad Laboratories, Hercules, CA, USA) and the images were analyzed using Image Lab software 6.1 (Molecular Imager Chemi Doc XRS+, Bio-Rad Laboratories, USA).

### 2.6. Statistical Analysis

The statistical evaluation was conducted using the Statistica CZ 12 software package version 4.2.2, with one-factor ANOVA followed by Tukey’s comparison test (*p* < 0.05). The data are presented as mean ± SD. To ensure normal distribution and homogeneity of variance, the data were verified using Levene’s test. General linear models were used to analyze the data and assess the effects of the pre-slaughter handling (anoxia and control) and time, as well as their interactions. A study was conducted to compare the storage efficiencies of rigor index, pH, hardness, and lipid oxidation (TBARS) after the death of fish. A three-way ANOVA was used to analyze the data using the R statistical program version 4.2.2. The study compared the effects of stocking densities (35, 50, and 65 kg·m^−3^) and pre-slaughter procedures (anoxia and control) on the aforementioned factors. For blood samples and antioxidant capacity enzyme analysis, a two-way ANOVA was conducted using the R statistical program version 4.2.2. The stocking densities (35, 50, and 65 kg·m^−3^) and pre-slaughter procedures (anoxia and control) were the two factors considered for the analysis. Before the ANOVA analysis, the normality and homogeneity of the data were checked. A three-way ANOVA *p*-value table for the data variables was included in Appendix A.

## 3. Results

### 3.1. pH

The pH in filets of the control groups (C) was markedly higher (*p* < 0.05) compared to anoxic (AN) filets regardless of fish stocking density (Figure 1). Furthermore, the pH decreased noticeably in both the control (C) and anoxic (AN) groups, by elapsing time. Moreover, pH was affected by the stocking density in both anoxic (AN) and control (C) groups during 72 h of refrigerator storage.

### 3.2. Rigor Mortis Index

The rigor index data following a 72-h stunning period and fish preservation at 4 °C are presented in Figure 2. Anoxia (AN) and control (C) groups had rigor mortis after 6 and 10 h, respectively. According to the results, rigor mortis started earlier in the fillets from fish of the anoxia groups (AN) compared to the control (C) without any effect of culture density. Furthermore, fish exposed to anoxia reached full rigor earlier than the control at all stocking densities. However, the development of the rigor index was unaffected by the fish stocking density.

### 3.3. Blood Glucose and Cortisol

Fish exposed to anoxia showed slightly higher plasma cortisol levels (Figure 3) but they were not significant. Moreover, stocking density did not show a significant impact on the cortisol level. Additionally, glucose concentration (Figure 4) was not affected by different stocking densities and its concentration was significantly higher in fish exposed to anoxia groups (AN) than in the control groups (C).

### 3.4. Antioxidant Enzymes Activity

Fish exposed to anoxia (AN) displayed significantly higher catalase (CAT) activity (Figure 5) compared to the control groups (C). Fish exposed to anoxia showed higher superoxide dismutase (SOD) activity than the control only at a stocking density of 65 kg·m^−3^ (Figure 6). At stocking densities of 50 and 65 kg·m^−3^, fish exposed to anoxia (AN) had significantly higher glutathione peroxidase (GPX) activity in comparison to the control group (C) (Figure 7). No significant effects were observed on CAT and GPX activities due to stocking density, while SOD activity showed a marked increase at stocking densities of 50 and 65 kg·m^−3^ in comparison to the lowest stocking density.

### 3.5. Hardness

The hardness of the control fillets (Group C) was significantly higher (*p* < 0.05) than the anoxic fillets (Group AN) after 48 h of storage (Figure 8). Furthermore, we observed a significant declining trend in all groups during the storage time. As the stocking density increased, the fillet hardness decreased gradually, especially in the anoxia (AN) group.

Regarding the effect of pre-slaughter handling, the control group showed a higher level of hardness compared to the anoxia group. This study confirmed that the control group had less protein and lipid oxidation development, as demonstrated by the Western blot and TBARs assay. Therefore, the higher oxidation development in the anoxia (AN) group could be related to the lower hardness.

### 3.6. Thiobarbituric Acid Reactive Substances (TBARS)

MDA concentration was significantly higher in the muscles of fish exposed to anoxia (AN) compared to the control groups (C) (Figure 9). Meanwhile, in fish exposed to anoxia (AN), the MDA concentration increased significantly after 12 h and then plateaued. Regarding the control group (C), the MDA concentration increased considerably after 24 h. Furthermore, MDA increased significantly during 48 h of storage at 4 °C in all treatments. The oxidation development was affected by different stocking densities in the fish exposed to anoxia (AN), where higher densities promoted lipid oxidation development.

### 3.7. Protein Changes during Postmortem

Several bands between 15 and 220 kDa were found in the protein profile of largemouth bass fillets (Figure 10A). The bands may be attributed to specific components such as myosin heavy chain (MHC) at 200 kDa, nebulin (107 kDa), actin (43 kDa), troponin (30 kDa), and myosin light chain components (25–15 kDa) [23]. Consistent protein patterns were observed among all groups except for the control group at 35 kg·m^−3^, which displayed an intense MHC on SDS-PAGE. The protein carbonyl groups with respect to Western blot (immunoblot) among all groups during 48 h storage at 4 °C (Figure 10B) showed an increasing trend in band intensity by elapsing time at approximately 43 kDa, which indicates actin protein degradation. Additionally, less oxidized carbonyl was observed in the control group (C) than in the group subjected to anoxia (AN). Furthermore, protein carbonyl oxidation was affected by density. Regardless of the stunning method, we observed a more intense band by increasing the stock density.

## 4. Discussion

Fish exposed to anoxia struggled more than control fish, according to our observations made during the slaughter period. This activity triggered earlier rigor mortis in anoxic fish as well as lower initial postmortem pH [24]. The result of our study is in accordance with the previous studies [25,26]. In contrast with our findings, ref. [10] did not observe any differences in pH during 3- and 6-min anoxia in Nile tilapias. Additionally, stress during slaughter can result in a higher ultimate pH or a faster pH decline after death [27]. The animal’s pre-slaughter reactions can change the muscle’s energy metabolism, significantly impacting the final muscle quality. For instance, lower pH and muscle energy are associated with exhaustion before slaughtering [28]. In addition, the reduction in muscle pH might be associated with the generation of H^+^ ions resulting from ATP degradation [29]. The pH decreased significantly until 24 h due to lactic acid formation and then, after 36 h, increased slightly owing to ammonia formation. Higher stocking density, on the other hand, is linked to more stress during rearing, which may cause lower levels of muscle glycogen and higher ultimate pH [27]. Fish subjected to high densities may exhibit increased struggle, leading to a faster depletion of glycogen stores. Glycogen plays a key role in fish muscle function, and its depletion can impact postmortem pH regulation. One possible reason for the observed phenomenon could be linked to the chronic stress experienced by the sampled fish. This stress could have been caused by various factors, including poor water quality and high stocking densities. Therefore, higher stocking density in both groups showed lower initial muscle pH compared to the lower density.

According to our results, the onset of rigor mortis started after 6 and 12 h in the fish exposed to anoxia and in the control, respectively. The results indicate that rigor mortis started earlier under anoxic conditions [30]. Ref. [31] revealed that in fish exposed to stressful conditions, rigor mortis starts earlier compared to unstressed fish. The delay of rigor mortis onset in control fish compared to anoxic fish depends on ATP decline. Because of limited glycogen reserves in fish muscle exposed to anoxia, postmortem analysis started earlier than that in the control group. Most likely, fish are exhausted when exposed to anoxic conditions, leading them to enter rigor mortis more rapidly [8,32]. The observed drop in pH, linked directly to the greater rigor mortis loss, suggests a clear correlation between these factors and the final fillet quality.

Assessment of fish stress commonly involves measuring cortisol levels and blood glucose concentration. When fish are stimulated, they produce hormones like cortisol, which increases the synthesis of glucose [33]. According to our results, exposing largemouth bass to anoxia led to raised blood cortisol levels. Moreover, a similar increasing tendency was observed for the plasma glucose level. This finding is in line with the results obtained in Channel catfish and Asian seabass [34,35]. An alteration in blood cortisol levels is considered the principal response to stress in the body [36]. The findings of our study indicated that regardless of stocking density, anoxia might induce stress in fish [36]. Cortisol activates the central nervous system by enhancing glucose synthesis via various metabolic pathways, such as the breakdown of muscle and liver glycogen reserves. In this context, anoxia has been observed to cause stress and physiological alterations in largemouth bass, as evidenced by changes in blood glucose and cortisol levels. During anoxia, fish undergo oxygen deprivation, leading to a metabolic shift from aerobic to anaerobic pathways. Fish resort to anaerobic glycolysis to produce energy since they cannot engage in aerobic respiration in the absence of oxygen. This process involves breaking down glycogen into glucose to produce ATP for cellular processes. As a result, blood glucose levels in fish can increase during anoxia since glycogen stores are mobilized to meet the energy demands of tissues [37,38]. As a result, the highest blood glucose level in the fish decreased, leading to a decrease in its utilization for biochemical reactions in the body after 3 min of pre-slaughter anoxia. Increased glucose production can assist fish in coping with the energy demands caused by stress. Fish release cortisol, a stress hormone, in response to various stressors, including anoxia. Cortisol helps regulate metabolism and energy mobilization during stress responses [37]. The fish reared at higher density may experience an increase in serum glucose and cortisol levels due to energy metabolism through gluconeogenesis and glycogenolysis pathways [38]. This augmentation in energy would help the fish cope with stressful situations, such as high stocking density. Nonetheless, raising stocking densities did not result in elevated blood cortisol and glucose levels, probably suggesting that fish in the current study acclimated to the density during the rearing period.

Rapid pH declines after death led to the build-up of muscle ROS and increased levels of antioxidant enzymes such as SOD, CAT, and GSH-Px [39]. Due to the activity of the antioxidant defense system during adaptation time, we observed that the SOD, CAT, and GPx activities increased under anoxic conditions in comparison to the control group. This condition implies that the antioxidant defense system may play a vital role in facilitating an adaptive mechanism to withstand anoxia. The activation of the antioxidant defense system may also be linked to the molecular mechanisms involved in oxygen detection and the associated transduction pathways that govern intermediate metabolism during anoxia [40].

Moreover, elevated stress levels during rearing were associated with increased stocking densities, leading to higher SOD activity at 50 and 65 kg·m^−3^ stocking densities compared to 35 kg·m−3 stocking densities. In agreement with our findings, [41] indicated higher SOD and MDA levels in fillets of blunt snout bream (*Megalobrama amblycephala*) reared at higher stocking density. Higher stocking density in aquaculture can lead to hypoxia [42], which triggers antioxidant defense. However, in our study stocking density did not affect CAT and GPX due to the lower level of antioxidant defense.

The reduction in hardness observed across all groups following storage is likely attributed to enzymatic protein degradation, as documented in prior studies [15,43,44]. Our study showed that lower stocking densities of 35 and 50 kg·m^−3^ produced firmer fillets compared to the higher density of 65 kg·m^−3^, suggesting that fish welfare during rearing impacts the ultimate quality of fillets [15]. However, ref. [45] reported no significant difference in hardness across all stocking densities. The present study found that the hardness decreased in all samples, which was consistent with previous research [46]. We hypothesize that fish reared under stocking densities of 35 and 50 (kg·m^−3^) may experience greater opportunities for vigorous swimming, a factor that could favorably influence the texture. Thus, increasing stocking density is contrary to fish welfare and has an adverse influence on fillet quality. According to [47], pre-slaughter stressors like crowding, noisy environments, and chasing led to an increase in the expression and activity of cathepsin B and L in farmed Atlantic salmon fillets. This increase in cathepsin activity can result in faster degradation of muscle proteins and negatively impact the texture of the fillets. Lower muscle hardness negatively impacts the economic value of fish by making them unsuitable for industrial processes or human consumption [48]. It has been observed that the fish stocked at higher densities displayed lower flesh quality. This occurred as a result of alterations in muscle structure, including a reduction in the area of muscular bundles and an increase in intramuscular connective tissue [34]. It is suggested that high stocking densities have negative effects on the quality of largemouth bass flesh.

In the current study, the consequence of anoxia was the acceleration of lipid oxidation, as confirmed by the elevated MDA concentration in the fillets of fish exposed to anoxia. Our speculation is that this could be associated with the excessive build-up of MDA. MDA can cause cytotoxicity and protein denaturation by cross-linking with proteins, nucleic acids, and amino phospholipids [49]. Fish exposed to anoxia demonstrated higher levels of lipid oxidation compared to the control group. In agreement with our result, ref. [50] reported a relationship between pre-slaughter stress and lipid oxidation in rainbow trout (*Oncorhynchus mykiss*). This could be associated with inadequate ATP production and a rapid decrease in pH levels [51,52]. Moreover, the higher pH level in anoxia fillets when compared to control fillets could serve as an alternate reason for reducing the formation of met-Hb and Hb-deoxygenation, while also minimizing the progression of lipid oxidation. Likewise, the higher lipid oxidation level under anoxic conditions in this experiment could be associated with the higher level of superoxide radicals or hydroxyl radicals.

Additionally, lipid oxidation development started earlier in anoxia than in the control group. In this regard, the results indicated that pre-slaughtering stressful situations can speed up the development of oxidation [53]. The higher presence of lipid-oxygenated products in the exposed fish fillet led to the progression of lipid oxidation during postmortem when compared to the control group. It might be one of the reasons that lipid oxidation started earlier in exposed fish than in the control group. Thus, severe stressful circumstances or stress duration may render the adaptive response ineffective, increasing lipid oxidation.

Moreover, lipid oxidation was affected by stocking density in the anoxia group. [41] found that fillets of blunt-snout bream (*Megalobrama amblycephala*) reared at higher stocking density had higher levels of SOD and MDA, which is consistent with our findings. Higher aquaculture stocking densities have the potential to induce hypoxia, which activates the antioxidant defense system [42].

During the refrigerator storage period, the protein patterns were analyzed in all groups by using SDS-PAGE to identify any protein alterations. We confirmed the impact of stocking density on protein profile by indicating MHC at a density of 35 kg·m^−3^ in the control group. Proteolysis is likely responsible for the susceptibility of MHC due to the formation of cross-linked proteins [54]. It appears that MHC was negatively impacted by high stocking density and anoxia in this study. According to [15], stocking density has a significant impact on the quality of African catfish fillets during storage. The SDS-PAGE result indicated the degradation of MHC in all groups except 35 kg·m^−3^.

Fish fillets exposed to anoxia had higher protein carbonyls than control fillets after 48 h of storage, as confirmed by Western blotting. The exposure of fish to anoxia resulted in higher protein carbonyls than control fillets. Protein carbonyls were more intense in anoxia fillets compared to control fillets during 48 h storage, which may be due to actin protein oxidation. Protein oxidation caused by metal-catalyzed cleavages is a significant form of oxidative damage during storage time [55]. During the postmortem period, protein degradation mainly occurs due to calpain activity [56]. Moreover, higher levels of lipid oxidation products can lead to increased protein oxidation (through metal catalyzation) in anoxia fillets, which may result in poorer textural parameters (hardness) compared to control fillets with less oxidized protein. It is reasonable to assume that even a slight degradation of actin can weaken the myofibrillar lattice, which can affect the texture of meat. The extent of the effect depends on where the thin filament’s actin is degraded. In agreement with our result, [15] reported higher protein oxidation confirmed by Western blot during postmortem storage of African catfish. We assume that at higher stocking densities, MDA can combine with the nucleophilic groups of proteins, leading to protein denaturation. On the other hand, when taking the stocking density into account, we noticed a lesser protein oxidation at lower stocking densities. According to [57], exercise results in more rigid connective tissue leading to a firmer texture. We propose that fish stocked at the density of 35 kg/m^−3^ may have greater opportunities for intense swimming, resulting in better texture and lower protein oxidation.

## 5. Conclusions

Recent research has confirmed that anoxia and stocking density significantly impact the ultimate fillet quality. Due to the increased stress from the fish exposed to anoxia, higher levels of lipid oxidation in their fillets were observed when compared to the control group. In this investigation, the influence of stress on the progression of protein oxidation was confirmed by Western blot analysis. In addition, the control group showed lower levels of antioxidant defense, blood cortisol, and glucose, as well as greater levels of hardness and pH than the anoxia group. We observed lower hardness and higher oxidation progress in the higher stocking density of the anoxic group. The present study confirmed the role of pre-slaughtering handling on the final fillet quality.

## Figures and Tables

**Figure 1 foods-13-01477-f001:**
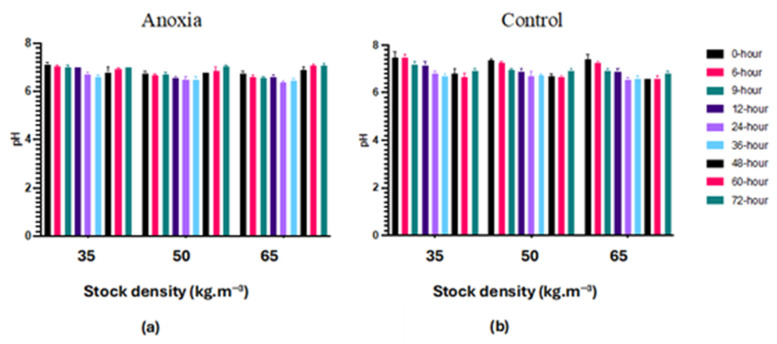
Changes of pH in largemouth bass fillet in anoxia (**a**) and control (**b**) groups during 72 h of refrigerated storage on ice by concerning three different fish stock densities (35, 50, and 65 kg·m^−3^) storage on ice (mean ± SD; *n* = 3).

**Figure 2 foods-13-01477-f002:**
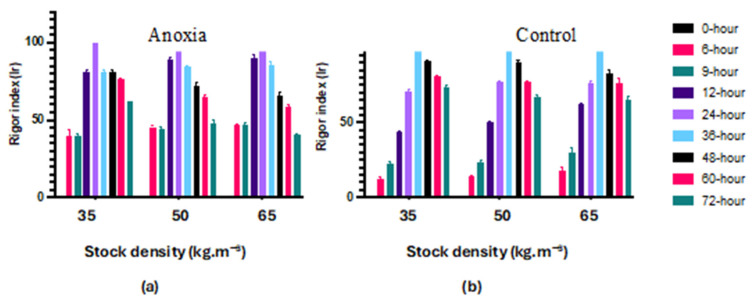
Changes of rigor index in largemouth bass fillets in anoxia (**a**) and control (**b**) groups during 72 h refrigerated storage on ice by concerning 3 different fish stock densities (35, 50 and 65 kg·m^−3^) (mean ± SD; *n* = 3).

**Figure 3 foods-13-01477-f003:**
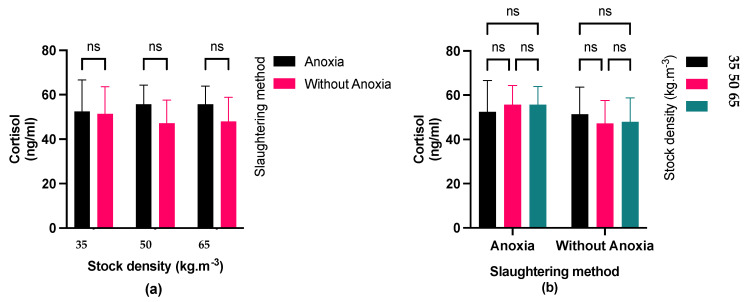
(**a**) Effect of the pre-slaughter procedures (anoxia and without anoxia) in the different fish densities (35, 50, and 65 kg·m^−3^) on cortisol levels during (*p* < 0.05) multiple comparisons test); (**b**) Effect of stocking densities (35, 50, and 65 kg·m^−3^) in different pre-slaughter procedures (anoxia and without anoxia groups) on cortisol level; (*p* < 0.05, Tukey’s multiple comparisons test). Bar graph showing mean and standard deviation (ns: non-significant).

**Figure 4 foods-13-01477-f004:**
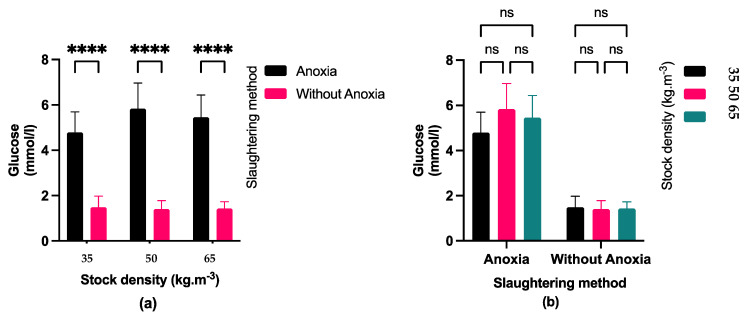
(**a**) Effect of pre-slaughter procedures (anoxia and without anoxia) in different stocking densities (35, 50, and 65 kg·m^−3^) on glucose level; (*p* < 0.05, multiple comparisons test); (**b**) Effect of stocking densities (35, 50 and 65 kg·m^−3^) in different pre-slaughter procedures (anoxia and without anoxia groups) on glucose level; (*p* < 0.05, Tukey’s multiple comparisons test). Bar graph showing mean and standard deviation (ns: non-significant; **** *p* < 0.0001).

**Figure 5 foods-13-01477-f005:**
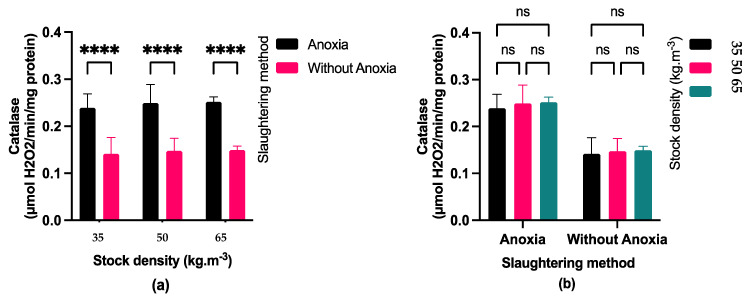
(**a**) Effect of pre-slaughter procedures (anoxia and without anoxia) under different stocking densities (35, 50 and 65 kg·m^−3^) on catalase activity; (*p* < 0.05, multiple comparisons test); (**b**) Effect of stocking densities (35, 50 and 65 kg·m^−3^) in different pre-slaughter procedures (anoxia and without anoxia groups) on catalase activity; (*p* < 0.05, Tukey’s multiple comparisons test). Bar graph showing mean and standard deviation (ns: non-significant; **** *p* < 0.0001).

**Figure 6 foods-13-01477-f006:**
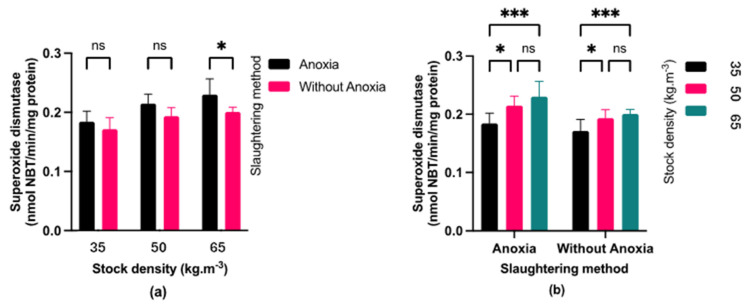
(**a**) Effect of pre-slaughter procedures (anoxia and without anoxia) under different stocking densities (35, 50, and 65 kg·m^−3^) on superoxide dismutase activity; (*p* < 0.05, multiple comparisons test); (**b**) Effect of stocking densities (35, 50 and 65 kg·m^−3^) in different pre-slaughter procedures (anoxia and without anoxia groups) on superoxide dismutase activity; (*p* < 0.05, Tukey’s multiple comparisons test). Bar graph showing mean and standard deviation (ns: non-significant; * *p* < 0.05; *** *p* < 0.001).

**Figure 7 foods-13-01477-f007:**
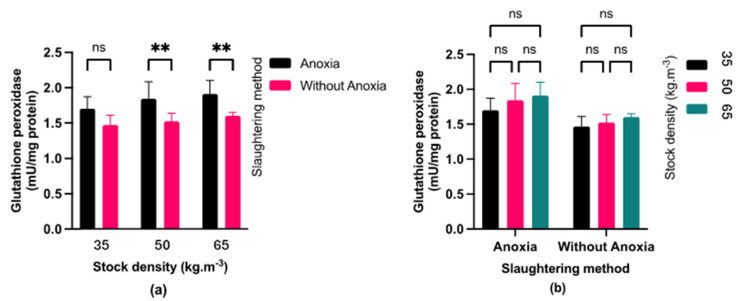
(**a**) Effect of pre-slaughter procedures (anoxia and without anoxia groups) on glutathione peroxidase activity during different stocking densities (35, 50 and 65 kg·m^−3^); (*p* < 0.05, multiple comparisons test); (**b**) Effect of stocking densities (35, 50 and 65 kg·m^−3^) on glutathione peroxidase activity in different pre-slaughter procedures (anoxia and without anoxia groups); (*p* < 0.05, Tukey’s multiple comparisons test). Bar graph showing mean and standard deviation (ns: non-significant; ** *p* < 0.01).

**Figure 8 foods-13-01477-f008:**
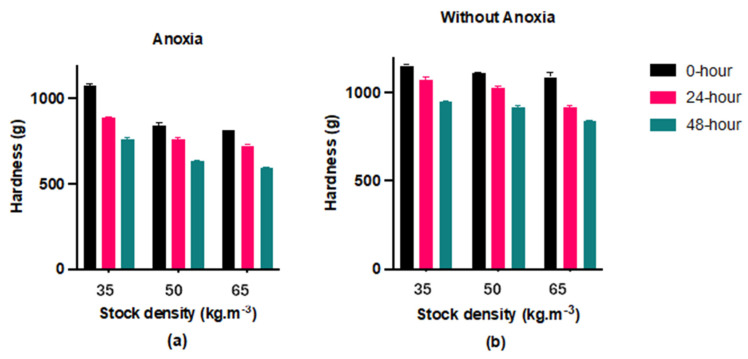
Hardness changes in largemouth bass fillet in anoxia (**a**) and without anoxia (**b**) groups during 48 h refrigerated storage at +4 °C with intervals 24 h by concerning three different fish stock densities (35, 50, and 65 kg·m^−3^) (mean ± S.D., *n* = 6).

**Figure 9 foods-13-01477-f009:**
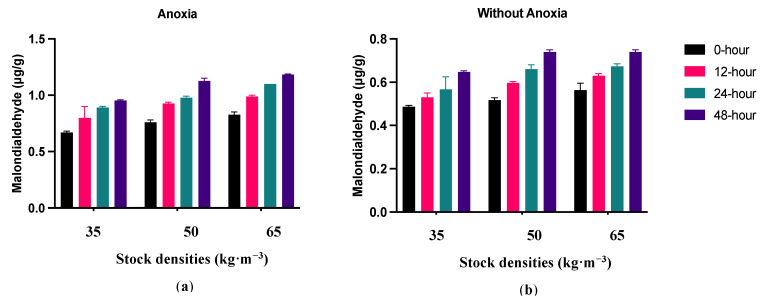
TBARS changes in largemouth bass fillet after anoxia (**a**) and without anoxia (**b**) groups during 48 h of refrigerated storage at +4 °C with by concerning three different fish stock densities (35, 50 and 65 kg·m^−3^) (mean ± S.D., *n* = 6).

**Figure 10 foods-13-01477-f010:**
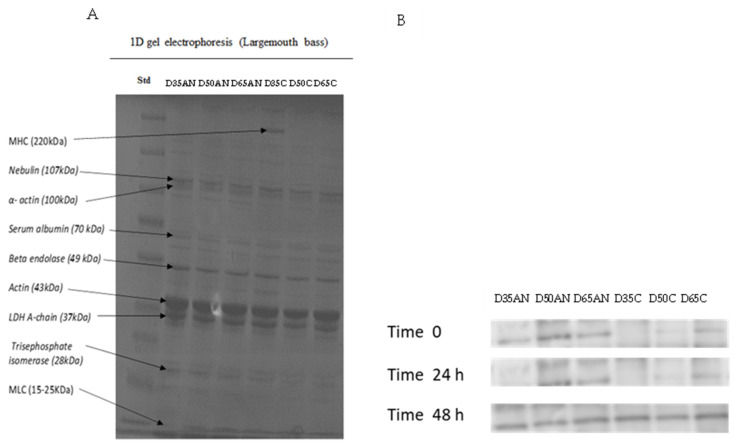
(**A**) SDS-polyacrylamide gel electrophoresis and (**B**) immunoblotting against protein carbonyl in largemouth bass (*Micropterus salmoides*) refrigerated storage fillet under +4 °C from fish groups cultured under three different fish densities, 35; 50; and 65 kg·m^−3^, subjected to anoxia and control conditions before the slaughtering. SD35C, SD50C, SD65C (C = Control), SD35AN, SD50AN, and SD65AN (AN = Anoxia).

## Data Availability

The original contributions presented in the study are included in the article and Appendix A, further inquiries can be directed to the corresponding author.

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
