# Peer review of "Effects of Stocking Density and Pre-Slaughter Handling on the Fillet Quality of Largemouth Bass (Micropterus salmoides): Implications for Fish Welfare"

_foods, 2024, doi:10.3390/foods13101477_

Round 1

Reviewer 1 Report

Comments and Suggestions for Authors

Here are some of my questions about this manuscript.

- Study Purpose: What is the main goal of the study on largemouth bass (Micropterus salmoides) and how does it relate to fish welfare and fillet quality?

- Pre-Slaughter Procedures: How do different pre-slaughter procedures affect the fillet quality of largemouth bass?

- Stocking Density: What impact does stocking density have on the welfare of largemouth bass and the quality of their fillets?

- Anoxia Group: What were the specific conditions and results observed in the anoxia group of largemouth bass?

- Control Group: How did the control group differ from the anoxia group in terms of pre-slaughter treatment and subsequent fillet quality?

- Antioxidant Enzyme Activity: What role do antioxidant enzymes play in the quality of largemouth bass fillets post-slaughter?

- Rigor Mortis Index: How was the rigor mortis index measured, and what were the findings regarding its development in the fish?

- Biochemical Analysis: What methods were used to analyze the blood biochemistry of the fish, and what stress indicators were measured?

- Proteomics: How were the muscle proteins of the largemouth bass analyzed, and what were the key findings?

- Fillet Hardness: What factors influenced the hardness of the largemouth bass fillets, and how did it change over time during storage?

Author Response

Response to Reviewer 1 Comments

Point 1: Study Purpose: What is the main goal of the study on largemouth bass (Micropterus salmoides) and how does it relate to fish welfare and fillet quality?

Response 1: Thank you for your constructive comment. We examined a comprehensive study on how fish welfare affects largemouth bass fillet quality during postmortem. Our study thoroughly examined two crucial factors (stunning method and stock density) in relation to fish welfare, aiming to identify correlations between the mentioned aspects of fish welfare and ultimate fillet quality. Additionally, the impact of the stunning method and stock density as an indicator of fish welfare on some quality parameters was investigated. In our current study, we also proposed the oxidation pathway mechanisms.

Point 2: Pre-Slaughter Procedures: How do different pre-slaughter procedures affect the fillet quality of largemouth bass?

Response 2: We appreciate your comment. We considered two aspects of fish welfare (stunning method and stock density) in this study to identify their impact on final fillet quality. The mentioned aspects induced stressful conditions to fish before stunning. We tried to show how fillet quality would be affected by the stressful conditions in the pre-slaughter procedure. For instance, oxidation progress and hardness were affected by the pre-slaughter procedure in our study.

Point 3: Stocking Density: What impact does stocking density have on the welfare of largemouth bass and the quality of their fillets?

Response 3: Thank you for your comment. The obtained results revealed a higher fillet quality in all aspects at lower stock density, compared to the higher one. Additionally, the fishery industry would like to produce more fish by increasing the stock density. However, higher stocking density in this study would not be appropriate for fish welfare. Furthermore, we suggest determining the optimal stock density to maximize fish production and fillet quality. We have already published one paper in this regard: Considering Two Aspects of Fish Welfare on African Catfish (Clarias gariepinus) Fillet throughout Postmortem Condition: Efficiency and Mechanisms.

Point 4: Anoxia Group: What were the specific conditions and results observed in the anoxia group of largemouth bass?

Response 4: We included more information in lines: 109-114. Experimental fish from each density group (18 fish per tank replicate) were randomly captured and subjected to an anoxia test by keeping the fish out of the water for 3 min. In this way, three experimental anoxia (AN) groups based on the different densities were created: D35AN, D50AN, and D65AN. In comparison to the control group, anoxia led to higher oxidation levels, reduced hardness, and lower pH. In addition, fish exposed to anoxia reached full rigor earlier than the control group. Anoxia resulted in higher levels of cortisol, glucose, and oxidative enzyme activity compared to the control group.

Point 5: Control Group: How did the control group differ from the anoxia group in terms of pre-slaughter treatment and subsequent fillet quality?

Response 5: We appreciate your comment. We explained in detail about this issue in lines: 115- 117 (Additionally, after purification, the same number of fish (n=54) from each stocking density group were killed by direct percussion on the head without air exposure. These three groups were considered control (C) groups (D35C, D50C, and D65C). Regarding the final fillet quality, the control group showed lower levels of antioxidant defense, blood cortisol, and glucose, as well as greater levels of hardness and pH than the anoxia group. In addition, the control group indicated lower development of protein oxidation which was confirmed by western blot analysis.

Point 6: Antioxidant Enzyme Activity: What role do antioxidant enzymes play in the quality of largemouth bass fillets post-slaughter?

Response 6: Thank you for your comment. We explained this issue in lines: 472-487. Due to impaired antioxidant defense mechanisms, ROS are not effectively removed from the body after fish death in the anoxia group. Antioxidant enzymes like SOD, CAT, and GSH-Px were found to have reduced activity in post-mortem at low pH in the anoxia group. Post-mortem fast pH decline (AN group) resulted in sarcoplasmic protein denaturation contributing to the accumulation of muscle ROS.

Point 7: Rigor Mortis Index: How was the rigor mortis index measured, and what were the findings regarding its development in the fish?

Response 7: Thank you for your comment.  The tail drop method was utilized to evaluate the onset of rigor mortis. We explain it in lines: 152-164. We reported that rigor mortis started earlier in the fillets from fish of the anoxia groups (AN) compared to the control (C). Furthermore, fish exposed to anoxia reached full rigor earlier than the control at all stocking densities. However, the development of the rigor index was unaffected by fish stocking density.

Point 8: Biochemical Analysis: What methods were used to analyze the blood biochemistry of the fish, and what stress indicators were measured?

Response 8: We describe in lines: 167-172 (After administering anesthesia with 0.03 ml/L clove oil, 18 fish, three from each group, had their caudal veins sampled for blood right after they were slaughtered and before they were filleted. This method is used to evaluate the stress levels of the fish by measuring the concentrations of glucose and cortisol. Heparinized syringes were used for this procedure). In the current study, we measured glucose and cortisol concentrations as an early stress indicator in the blood fish.

Point 9: Proteomics: How were the muscle proteins of the largemouth bass analyzed, and what were the key findings?

Response 9: We explained this issue in lines: 216-220 (Almost 100 mg of frozen fish muscle from six right fillets in each group was cut and weighed at -20°C.  The muscle tissue was ground in 50 mM phosphate buffered saline (PBS) solution (500 µl). This solution was prepared with a 0.01 M phosphate buffer concentration and a 0.154 M sodium chloride concentration at pH 7.4. The crude extracts were then transferred to an Eppendorf tube).

We included this information in lines: (377-383). Consistent protein patterns were observed among all groups except for the control group at 35 kg.m-3, which displayed an intense MHC on SDS-PAGE. Western blot (immunoblot) among all groups showed an increasing trend in band intensity by elapsing time at approximately 43 kDa, which indicates actin protein degradation. Additionally, less oxidized carbonyl was observed in the control group (C) than in the group subjected to anoxia (AN). Furthermore, protein carbonyl oxidation was affected by density. Regardless of the stunning method, we observed a more intense band by increasing the stock density.

Point 10: Fillet Hardness: What factors influenced the hardness of the largemouth bass fillets, and how did it change over time during storage?

Response 10: In the current study, the consequence of anoxia revealed lower hardness in the AN group due to the higher protein degradation. It seems that protein degradation influenced hardness rather than other factors. Additionally, hardness decreased in all groups during the storage time.

Reviewer 2 Report

Comments and Suggestions for Authors

In this study, the effect of pre-slaughter methods of percussion on the head and exposure for 3 minutes and the stocking densities on fish quality were observed. It is useful for fish quality improvement and fish welfare. There needs to do some revision.

1.     In the abstract, there should be presented some key result numbers.

2.     In the introduction section, there is very little introduction for the pre-slaughter. Why this study to study the anoxia using “exposure for 3 minutes AN treatments”. What is innovation for this pre-slaughter.

3.     In lines 69, there are two words “in”.

4.     Fishes were cut two fillets. Let- and right-hand fillets were used to analyze different quality index. But in the same fillets, different position may have different lipid and protein content, so the position for each index should given.

5.     What is the role of abbreviate of “AN” and “C”. for example, in Figure 1-2, Changes of rigor index in largemouth bass fillets after anoxia (a) and without anoxia (control) (b). there is no the abbreviate. And in the text, authors also use the word “Anoxia (AN) and control (C)” together.

6.     In figure 2, the standard deviation of rigor index were not shown when it was 100 Ir. But the unit is above 100. So it can show the standard deviation.

7.     Check figure 3-6 there are different stock density 30-45-60.

8.     In figure 9, abscissa is not storage time.

9.     In the discussion section, relationship between each index should be discussed deeply.

Comments on the Quality of English Language

it can be improved.

Author Response

Response to Reviewer 2 Comments

Point 1:   In the abstract, there should be presented some key result numbers.

Response 1: We included in lines: 24-27.

Point 2: In the introduction section, there is very little introduction for the pre-slaughter. Why this study to study the anoxia using “exposure for 3 minutes AN treatments”. What is innovation for this pre-slaughter.

Response 2: Thank you for your comment. We included more information in lines: 47-50.

Point 3: In lines 69, there are two words “in”.

Response 3: We corrected in line 75.

Point 4: Fishes were cut two fillets. Let- and right-hand fillets were used to analyze different quality index. But in the same fillets, different position may have different lipid and protein content, so the position for each index should given.

Response 4: Thank you for your constructive comment. We would like to mention that we homogenized all the fillets before using them for chemical analysis to have almost the same quality. Regarding the textural parameter we used 3 parts of each fillet in all groups with 6 replications to minimize the error.

Point 5: What is the role of abbreviate of “AN” and “C”. for example, in Figure 1-2, Changes of rigor index in largemouth bass fillets after anoxia (a) and without anoxia (control) (b). there is no the abbreviate. And in the text, authors also use the word “Anoxia (AN) and control (C)” together.

Response 5: Yes, we agree. We corrected all figures accordingly.

Point 6: In figure 2, the standard deviation of rigor index were not shown when it was 100 Ir. But the unit is above 100. So it can show the standard deviation.

Response 6: We appreciate your comment. We would like to inform you that there was no statistical different we did not include it.

Point 7: Check figure 3-6 there are different stock density 30-45-60.

Response 7: Yes, we corrected.

Point 8:   In figure 9, abscissa is not storage time.

Response 8: Thank you for your comment. We corrected.

Point 9: In the discussion section, relationship between each index should be discussed deeply.

Response 9: Thank you for your constructive comment. We included more details in lines: 429-431, 440-442, 460-463, 509-512, and 555-557.

Comments on the Quality of English Language: it can be improved. Thank you for your suggestion. We would like to inform you that we sent our manuscript to the Elsevier English Editing Service (Invoice number LSD1000018695) for English correction.
